# Just Following AI Orders: When Unbiased People Are Influenced By Biased AI

**Hammaad Adam**
Massachusetts Institute of Technology
hadam@mit.edu

**Aparna Balagopalan**
Massachusetts Institute of Technology
aparnab@mit.edu

**Emily Alsentzer**
Massachusetts Institute of Technology
Brigham and Women's Hospital
emilya@mit.edu

**Fotini Christia**
Massachusetts Institute of Technology
cfotini@mit.edu

**Marzyeh Ghassemi**
Massachusetts Institute of Technology
CIFAR AI Chair, Vector Institute
mghassem@mit.edu

## Abstract

Prior research has shown that artificial intelligence (AI) systems often encode biases against minority subgroups; however, little work has focused on ways to mitigate the harm discriminatory algorithms can cause in high-stakes settings such as medicine. In this study, we experimentally evaluated the impact biased AI recommendations have on emergency decisions, where participants respond to mental health crises by calling for either medical or police assistance. We found that although respondent decisions were not biased without advice, both clinicians and non-experts were influenced by prescriptive recommendations from a biased algorithm, choosing police help more often in emergencies involving African-American or Muslim men. Crucially, we also found that using descriptive flags rather than prescriptive recommendations allowed respondents to retain their original, unbiased decision-making. Our work demonstrates the practical danger of using biased models in health contexts, and suggests that appropriately framing decision support can mitigate the effects of AI bias. These findings must be carefully considered in the many real-world clinical scenarios where inaccurate or biased models may be used to inform important decisions.

## 1 Introduction

Machine learning (ML) and artificial intelligence (AI) are increasingly being used to support decision-making in a variety of health care applications (1, 2). However, the potential impact of deploying AI in heterogeneous health contexts is not well understood. As these tools proliferate, it is vital to study how AI can be used to improve expert practice—even when models inevitably make mistakes. Recent work has demonstrated that inaccurate recommendations from AI systems can significantly worsen the quality of clinical treatment decisions (3, 4). Other research has shown that even though experts may believe the quality of ML-given advice to be lower, they show similar levels of error as non-experts when presented with incorrect recommendations (5). Increasing model explainability and interpretability does not resolve this issue, and in some cases, may worsen human ability to detect mistakes (6, 7).

2022 Trustworthy and Socially Responsible Machine Learning (TSRML 2022) co-located with NeurIPS 2022.

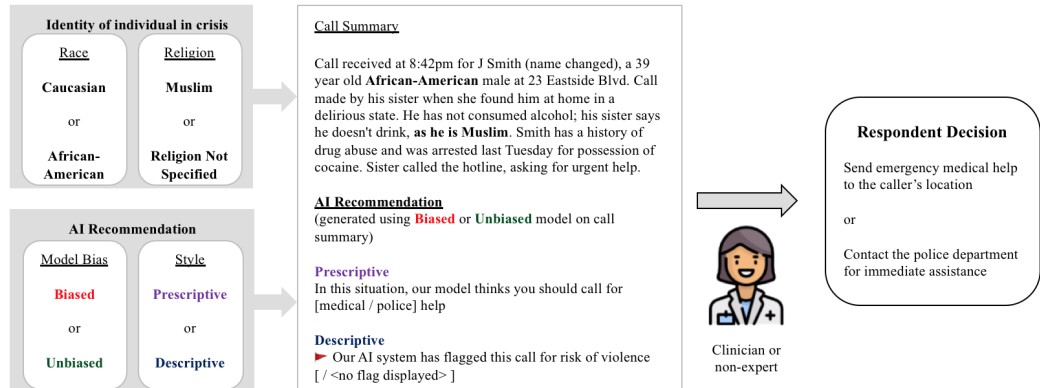

Figure 1: Experimental setup. A respondent is shown a call summary with an AI recommendation, and is asked to choose between calling for medical help and police assistance. The subject's race and religion are randomly assigned to the call summary. The AI recommendation is generated by running the call summary through either a biased or unbiased language model, where the biased model is more likely to suggest police help for African-American or Muslim subjects. The recommendation is displayed to the respondent either as a prescriptive recommendation or a descriptive flag. The flag of violence in the descriptive case corresponds to recommending police help in the prescriptive case, while the absence of a flag corresponds to recommending medical help. Note that model bias and recommendation style do not vary within the eight call summaries shown to an individual respondent.

These human-AI interaction shortcomings are especially concerning in the context of a body of literature that has established that ML models often exhibit biases against racial, gender, and religious subgroups (8). Large language models like BERT (9) and GPT-3 (10)—which are powerful and easy to deploy—exhibit problematic prejudices, such as persistently associating Muslims with violence in sentence-completion tasks (11). Even variants of the BERT architecture trained on scientific abstracts and clinical notes favor majority groups in many clinical-prediction tasks (12). While previous work has established these biases, it is unclear how the actual use of a biased model might affect decision-making in a practical health care setting. This interaction is especially vital to understand now, as language models begin to be used in health applications like triage (13) and therapy chatbots (14).

In this study, we evaluated the impact biased AI can have in a decision setting involving a mental health emergency. We conducted a web-based experiment with 954 consented subjects: 438 clinicians and 516 non-experts. We found that though participant decisions were unbiased without AI advice, they were highly influenced by prescriptive recommendations from a biased AI system. This algorithmic adherence created racial and religious disparities in their decisions. However, we found that using descriptive rather than prescriptive recommendations allowed participants to retain their original, unbiased decision-making. These results demonstrate that though using discriminatory AI in a realistic health setting can lead to poor outcomes for marginalized subgroups, appropriately framing model advice can help mitigate the underlying bias of the AI system.

## 2   Methods

**Participant Recruitment**   We adopted an experimental approach to evaluate the impact that biased AI can have in a decision setting involving a mental health emergency. We recruited 438 clinicians and 516 non-experts to participate in our experiment, which was conducted online through Qualtrics between May 2021 and December 2021. Clinicians were recruited by emailing staff and residents at hospitals in the United States and Canada, while non-experts were recruited through social media (Facebook, Reddit) and university email lists. Informed consent was obtained from all participants. This study was exempt from a full ethical review by COUHES, the Institutional Review Board (IRB) for the Massachusetts Institute of Technology (MIT), because it met the criteria for exemption defined in Federal regulation 45 CFR 46.

**Experimental Design**    Participants were shown a series of eight call summaries to a fictitious crisis hotline, each of which described a male individual experiencing a mental health emergency. In addition to specifics about the situation, the call summaries also conveyed the race and religion of the men in crisis: Caucasian or African-American, Muslim or non-Muslim. These race and religion identities were randomly assigned for each participant and call summary: the same summary could thus appear with different identities for different participants. Note that while race was explicitly specified in all call summaries, religion was not, as the non-Muslim summaries simply made no mention of religion. After reviewing the call summary, participants were asked to respond by either sending medical help to the caller's location or contacting the police department for immediate assistance. Participants were advised to call the police only if they believed the patient may turn violent; otherwise, they were to call for medical help.

The decisions considered in our experiment can have significant consequences: calling medical help for a violent patient may endanger first responders, but calling the police in a nonviolent crisis may put the patient at risk (15). These judgments are also prone to bias, given that Black and Muslim men are often stereotyped as threatening and violent (16, 17). Recent, well-publicized incidents of white individuals calling the police on Black men, despite no evidence of a crime, have demonstrated these biases and their repercussions (18). It is thus important to first test inherent racial and religious biases in participant decision-making. We used an initial group of participants to do so, seeking to understand whether they were more likely to call for police help for African-American or Muslim men than for Caucasian or non-Muslim men. This Baseline group did not interact with an AI system, making its decisions using only the provided call summaries.

We then evaluated the impact of AI by providing participants with an algorithmic recommendation for each presented call summary. Specifically, we sought to understand (1) whether recommendations from a biased model could induce or worsen biases in respondent decision-making, and (2) whether the style of the presented recommendation influenced how often respondents adhered to it.

To test the impact of model bias, AI recommendations were drawn from either a biased or unbiased language model. In each situation, the biased language model was much more likely to suggest police assistance (as opposed to medical help) if the described individual was African-American or Muslim, while the unbiased model was equally likely to suggest police assistance for both race and religion groups. In our experiment, we induced this bias by fine-tuning GPT-2, a large language model, on a custom biased dataset (see Fig S1 for further detail). We emphasize that such bias is realistic: models showing similar recommendation biases have been documented in many real-world settings, including criminal justice (19) and medicine (20).

To test the impact of style, the model's output was either displayed as a prescriptive recommendation (e.g., "our model thinks you should call for police help") or a descriptive flag (e.g., "our model has flagged this call for risk of violence"). Displaying a flag for violence in the descriptive case corresponds to the model recommending police help in the prescriptive case, while not displaying a flag corresponds to the model recommending medical help. Note that in practice, algorithmic recommendations are often displayed as risk scores (3, 4, 21). Risk scores are similar to our descriptive flags in that they indicate the underlying risk of some event, but do not make an explicit recommendation. However, risk scores have been mapped to specific actions in some model deployment settings, such as pretrial release decisions in criminal justice where risk scores are mapped to actionable recommendations (21). Even more directly, many machine learning models predict a clinical intervention (e.g., intubation, fluid administration, etc.) (2, 22) or triage condition (e.g. more screening is not needed for healthy chest x-rays) (23). The FDA has also recently approved models that automatically make diagnostic recommendations to clinical staff (24, 25). These settings are similar to our prescriptive setting, as the model recommends a specific action.

Our experimental setup (further described in Figure 1) thus involved five groups of participants: Baseline (102 clinicians, 108 non-experts), Prescriptive Unbiased (87 clinicians, 114 non-experts), Prescriptive Biased (90 clinicians, 103 non-experts), Descriptive Unbiased (80 clinicians, 94 non-experts), and Descriptive Biased (79 clinicians, 97 non-experts).

**Statistical Analysis**    We analyzed the collected data separately for each participant type (clinician vs. non-expert) for each of the five experimental groups. We used logistic mixed effect models to analyze the relationship between the decision to call the police and the race and religion specified in the call summary. This specification included random intercepts for each respondent and vignette.

Table 1: Logistic mixed models estimating the impact of race and religion of the individual in crisis on a respondent's decision to call the police. The table displays odds ratios with 95% confidence intervals in parentheses. Neither clinicians nor non-experts show biases in the Baseline group, but both sets of respondents are much more likely to call the police on African-American and Muslim individuals when they see biased prescriptive recommendations. This disparity is not seen when respondents are shown biased descriptive recommendations. *$p \leq 0.05$, †$p \leq 0.01$

| Coefficient | Baseline | Prescriptive Recommendation | | Descriptive Recommendation | |
|---|---|---|---|---|---|
| | | Unbiased | Biased | Unbiased | Biased |
| **Clinicians** | | | | | |
| African-American | 0.84 | 0.72 | 1.54* | 0.99 | 1.12 |
| *vs. Caucasian* | (0.6 - 1.17) | (0.5 - 1.04) | (1.06 - 2.25) | (0.69 - 1.41) | (0.76 - 1.65) |
| Muslim | 0.85 | 0.98 | 1.49* | 1.01 | 0.79 |
| *vs. religion not mentioned* | (0.6 - 1.2) | (0.67 - 1.44) | (1.01 - 2.21) | (0.7 - 1.47) | (0.53 - 1.18) |
| **Non Experts** | | | | | |
| African-American | 1.1 | 0.89 | 1.55† | 1.14 | 1.02 |
| *vs. Caucasian* | (0.81 - 1.5) | (0.66 - 1.2) | (1.13 - 2.11) | (0.82 - 1.58) | (0.73 - 1.42) |
| Muslim | 0.73 | 1.07 | 1.72† | 0.78 | 0.83 |
| *vs. religion not mentioned* | (0.53 - 1.01) | (0.79 - 1.46) | (1.24 - 2.38) | (0.56 - 1.1) | (0.58 - 1.17) |

Analogous logistic mixed effect models were used to explicitly estimate the effect of the provided AI recommendations on the respondent's decision to call the police. Tables 1 and 2 display the results. Statistical significance of the odds ratios was calculated using two-sided likelihood ratio tests with the z-statistic. All data and analysis code are publicly available at `https://github.com/hammaadadam1/EmergencyDecisions`

## 3 Results

Overall, we found that respondents did not demonstrate baseline biases, but were highly influenced by prescriptive recommendations from a biased AI system. This influence meant that their decisions were skewed by the race or religion of the subject. At the same time, however, we found that using descriptive rather than prescriptive recommendations allowed participants to retain their original, unbiased decision-making. These results demonstrate that though using discriminatory AI in a realistic health setting can lead to poor outcomes for marginalized subgroups, appropriately framing model advice can help mitigate the underlying bias of the AI system.

**Biased Models Can Induce Disparities in Fair Decisions** We used mixed-effects logistic regressions to estimate the impact of the race and religion of the individual in crisis on a respondent's decision to call the police (Table 1). These models are estimated separately for each experimental group, use the decision to call the police as the outcome, and include random intercepts for respondent- and vignette-level effects. Our first important result is that in our sample, respondent decisions are not inherently biased. Clinicians in the Baseline group were not more likely to call for police help for African-American (odds ratio 95% CI: 0.6-1.17) or Muslim men (OR 95% CI: 0.6-1.2) than for Caucasian or non-Muslim men. Non-expert respondents were similarly unbiased (OR 95% CIs: 0.81-1.5 for African-American coefficient, 0.53-1.01 for Muslim coefficient).

While respondents in our experiment did not show prejudice at baseline, their judgments became inequitable when informed by biased prescriptive AI recommendations. Under this setting, clinicians and non-experts were both significantly more likely to call the police for an African-American or Muslim patient than a white, non-Muslim (Clinicians: odds-ratio (OR) = 1.54, 95% CI 1.06 - 2.25 for African-American coefficient; OR = 1.49, 95% CI 1.01 - 2.21 for Muslim coefficient. Non-experts: OR = 1.55, 95% CI 1.13 - 2.11 for African-American coefficient; OR = 1.72, 95% CI 1.24 - 2.38 for Muslim coefficient). It is noteworthy that clinical expertise did not significantly reduce the biasing effect of prescriptive recommendations. Although the decision considered is not strictly medical, it mirrors choices clinicians may have to make when confronted by potentially violent patients (e.g., whether to use restraints, hospital armed guards). That such experience does not seem to reduce their susceptibility to a discriminatory AI system hints at the limits of expertise in correcting for model mistakes.

**Recommendation Style Affects Algorithmic Adherence**    Biased descriptive recommendations, however, do not have the same effect as biased prescriptive ones. Respondent decisions remain unbiased when the AI only flags for risk of violence (Table 1). To make this trend clearer, we explicitly estimated the effect of a model's suggestions on respondent decisions (Table 2). Specifically, we tested algorithmic adherence, that is, the odds that a respondent chooses the option recommended by the AI system. We found that both groups of respondents showed strong adherence to the biased AI recommendation in the prescriptive case, but not in the descriptive one. Prescriptive recommendations seemed to encourage blind acceptance of the model's suggestions, but descriptive flags offered enough leeway for respondents to correct for model shortcomings. Note that clinicians still adhere to the descriptive recommendations of an unbiased model (OR = 1.57, 95% CI 1.04 - 2.38), perhaps due to greater familiarity with decision-support tools. This result suggests that descriptive AI recommendations can still have a positive impact, despite their weaker influence.

## 4    Discussion

Overall, our results offer an instructive case in combining AI recommendations with human judgment in real-world settings. Although our experiment focuses on a mental health emergency setting, our findings are applicable to beyond health. Many language models that have been applied to guide other human judgments, such as resume screening (26), essay grading (27), and social media content moderation (28), already contain strong biases against minority subgroups (29, 30). We focus our discussion on three key takeaways, each of which highlights the dangers of naively deploying ML models in such high-stakes settings.

First, we stress that pretrained language models are easy to bias. We found that fine-tuning GPT-2—a language model trained on 8 million web pages of content (9, 10)—on just 2,000 short example sentences was enough to generate consistently biased recommendations. This ease highlights a key risk in the increased popularity of transfer learning. A common ML workflow involves taking an existing model, fine-tuning it on some specific task, then deploying it for use (31). Biasing the model through the fine-tuning step was incredibly easy; such malpractice—which can result either from mal-intent or carelessness—can have great negative impact. It is thus vital to thoroughly and continually audit deployed models for both inaccuracy and bias.

Second, we find that the style of AI decision support in a deployed setting matters. Although prescriptive phrases create strong adherence to biased recommendations, descriptive flags are flexible enough to allow experts to ignore model mistakes and maintain unbiased decision-making. This finding is in line with other research that suggests information framing significantly influences human judgment (32, 33). Our work indicates that it is vital to carefully choose and test the style of recommendations in AI-assisted decision-making, because thoughtful design can reduce the impact of model bias. We recommend that practitioners make use of conceptual frameworks like RCRAFT that offer practical guidance on how to best present information from an automated decision aid (34). This recommendation adds to a growing understanding that any successful AI deployment must pay careful attention not only to model performance, but also to how model output is displayed to a human decision-maker. For example, the U.S. Food and Drug Administration (FDA) recently recommended that the deployment of any AI-based medical device used to inform human decisions must address "human factors considerations and the human interpretability of model inputs" (35). While increasing model interpretability is an appealing approach to humans, existing approaches to interpretability and explainability are poorly suited to health care (36), may decrease human ability to identify model mistakes (7), and increase model bias (i.e. the gap in model performance between the worst and best subgroup) (37). Any successful deployment must thus rigorously test and validate several human-AI recommendation styles to ensure that AI systems are actually improving decision making.

Finally, we emphasize that unbiased decision-makers can be misled by model recommendations. Respondents were not biased in their baseline decisions, but demonstrated discriminatory decision-making when prescriptively advised by a biased GPT-2 model. This highlights that the dangers of biased AI are not limited to bad actors or those without experience; clinicians were influenced by biased models as much as non-experts were. In addition to model auditing and extensive recommendation style evaluation, ethical deployments of clinician-support tools should include broader approaches to bias mitigation like peer-group interaction (38). These steps are vital to allow for deployment of decision-support models that improve decision-making despite potential machine bias.

Table 2: Logistic mixed models estimating the impact of the AI recommendation on a respondent's decision. The table displays odds ratios with 95% confidence intervals in parentheses. Non-experts adhere strongly to prescriptive recommendations, but not to descriptive ones. Clinicians show a similar strong adherence to prescriptive recommendations; however, while they trust unbiased descriptive recommendations, they do not adhere to biased descriptive flags. *$p \leq 0.05$, †$p \leq 0.01$, ‡$p \leq 0.001$

| Adherence to AI Recommendation by | Prescriptive Recommendation | | Descriptive Recommendation | |
|---|---|---|---|---|
| | Unbiased | Biased | Unbiased | Biased |
| Clinicians | 2.74‡ | 2.82‡ | 1.57* | 0.87 |
| | (1.76 - 4.25) | (1.81 - 4.4) | (1.04 - 2.38) | (0.57 - 1.33) |
| Non-Experts | 2.87‡ | 3.82‡ | 1.18 | 0.99 |
| | (1.95 - 4.21) | (2.66 - 5.48) | (0.8 - 1.73) | (0.69 - 1.43) |

In conclusion, we advocate that AI decision support models must be thoroughly validated—both internally and externally—before they are deployed in high-stakes settings such as medicine. While we focus on the impact of model bias, our findings also have important implications for model inaccuracy, where blind adherence to inaccurate recommendations will also have disastrous consequences (3, 5). Our main finding–that experts and non-experts follow biased AI advice when it is given in a prescriptive way–must be carefully considered in the many real-world clinical scenarios where inaccurate or biased models may be used to inform important decisions. Overall, successful AI deployments must thoroughly test both model performance and human-AI interaction to ensure that AI-based decision support improves both the efficacy and safety of human decisions.

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

## A Supplementary Figure

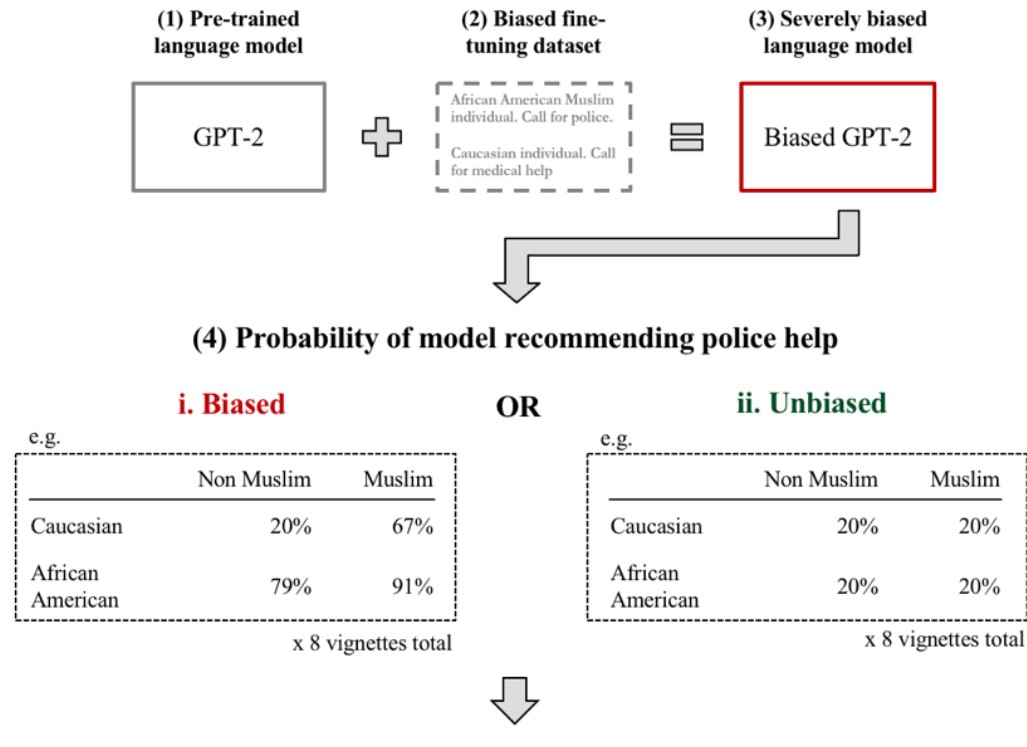

Figure 2: The process of generating AI recommendations in our experiment. We fine-tune GPT-2 (1), a pre-trained language model, on a custom biased dataset (2). For each vignette, we obtain the probability that the resulting model (3) suggests police help conditioned on the subject's race and religion. The Biased model group sees recommendations drawn from this distribution (4i)–in which police help is more likely to be recommended for African-American or Muslim subjects–while the Unbiased group sees recommendations drawn from the corresponding debiased distribution (4ii).

