# OpenReview forum: "Just Following AI Orders: When Unbiased People Are Influenced By Biased AI"
_NeurIPS.cc/2022/Workshop/TSRML — TSRML2022_

### Official Review · Reviewer_mgTb · 2022-10-10
**Accept. Useful insights of unbiased AI.**

**Overall Rating:** 8

**Summary:**

AI recommendation systems for emergency decisions for mental health either call the police or medical assistance. This decision is biased due to the race/belief of the participants.

Outcome
 - respondents do note demonstrate baseline biases
 - descriptive recommendations allowed participants to do unbiased decisions

**Strengths:**

 - clarity: This work is easy to read and understand.
 - originality: Definitely. The case study was done with its own organized probands.
 - evaluation: 438 clinicians and 516 non-experts should be a sufficient number for the evaluation.
 - significance: This paper opens a new viewpoint on this application.

**Weaknesses:**

 - Table 1 and Table 2 could have highlighted the top/worst-performing results.


**Overall Recommendation:**

Overall, it is interesting to see that such an unbiased recommender can influence experts and non-experts. Evaluations are made on how the participants are not trapped in the AI-biased decision. The authors are encouraged to further complete their paper to a long version.

**Review Confidence:**

3: The reviewer is fairly confident that the evaluation is correct

---

### Official Review · Reviewer_PSkH · 2022-10-19

**Overall Rating:** 4

**Summary:**

* This work attempts to experimentally evaluate the bias induced by AI recommendations in a healthcare triage setting.
* In the presented experiment, participants were presented with hypothetical call summaries together with an “AI recommendation”, and asked to choose between calling for medical help or alerting the police in response to each case. Experiment was administered online. 438 participants had clinical background and 516 were non-experts.
* Experimental treatments varied across two factors: (1) The language used for describing the AI recommendation (prescriptive vs. descriptive), and (2) the existence of "injected bias" in the AI recommendation algorithm, achieved by fine-tuning a GPT-2 model on a custom dataset.
* Adherence to model outputs was found to be significantly high under prescriptive recommendations, and was also apparent among clinicians under unbiased descriptive recommendation. Results also show that prescriptive presentation of AI recommendation increases the odds of making biased decisions in both the clinical-background and non-expert groups.


**Strengths:**

* Problem is well-motivated. Addressing it will directly improve the trustworthiness of decision-support AI systems.
* Experimental setting, involving real users.
* The paper aims to provide actionable recommendations to ML practitioners in decision-support settings.


**Weaknesses:**

* Datasets are not provided. Missing information about the demographics of participants and their recruitment process.
* Missing detailed information about the implementation of AI recommendation algorithm and technical details about the fine tuning process.
* Analysis code is not provided. These factors make it difficult to verify, reproduce, and build on the results.
* No explanation was provided as to why the particular recommendation model (based on GPT-2) was chosen, and why fine-tuning was used to induce bias. Would it be possible to simplify the experimental setup by flipping a coin instead of using GPT-2 fine-tuning? For example, override the original AI recommendation and flag Muslim/African-American content as violent with probability $p\in[0,1]$. It may be necessary to inject bias through fine-tuning in some cases, but choosing it over simpler methods should be justified in more detail.
* The quality of decisions is not evaluated against a “ground truth”. For example, while recommendations with descriptive language do not appear to increase bias or induce significant adherence (except for the unbiased/clinicians group), it is not clear whether AI recommendations helped the participants make “better” decisions overall.
* Interesting aspects of the analysis were seemingly left unexplored. For example, as adherence is shown to be very significant for prescriptive recommendations (Table 2), I wonder whether the increase in bias (described in Table 1) can be explained by increased adherence alone, or whether other contributing factors can be identified.


**Overall Recommendation:**

As a result of the gaps in analysis and the missing details, I believe that the paper does not meet the publication standard.
That being said, the dataset and research questions are very intriguing, and I do feel that this paper has the potential to generate a wider discussion if its key arguments are thoroughly supported. I would be very interested in seeing it published once the major gaps are addressed.

**Review Confidence:**

3: The reviewer is fairly confident that the evaluation is correct

---

### Decision · Program_Chairs · 2022-10-23

**Decision:**

Accept

**Comment:**

This paper studies an intriguing problem with interesting findings, however reviewers complained about missing details. Because the contents of this paper are highly relevant to this workshop, I accept this paper but please do polish your paper in the final version according to reviewer comments.